# View Gap Matters: Cross-view Topology and Information Decoupling for Multi-view Clustering

## ABSTRACT

Multi-view clustering, a pivotal technology in multimedia research, aims to leverage complementary information from diverse perspectives to enhance clustering performance. The current multi-view clustering methods normally enforce the reduction of distances between any pair of views, overlooking the heterogeneity between views, thereby sacrificing the diverse and valuable insights inherent in multi-view data. In this paper, we propose a **T**ree-Based View-**G**ap **M**aintaining **M**ulti-View **C**lustering (**TGM-MVC**) method. Our approach introduces a novel conceptualization of multiple views as a graph structure. In this structure, each view corresponds to a node, with the view gap, calculated by the cosine distance between views, acting as the edge. Through graph pruning, we derive the minimum spanning tree of the views, reflecting the neighbouring relationships among them. Specifically, we applied a share-specific learning framework, and generate view trees for both view-shared and view-specific information. Concerning shared information, we only narrow the distance between adjacent views, while for specific information, we maintain the view gap between neighboring views. Theoretical analysis highlights the risks of eliminating the view gap, and comprehensive experiments validate the efficacy of our proposed TGM-MVC method.

## CCS CONCEPTS

• **Theory of computation → Unsupervised learning and clustering**; • **Computing methodologies → Cluster analysis**.

## KEYWORDS

multi-view clustering, view gap

## 1 INTRODUCTION

In real world scenarios, we often face the challenge of learning from multiple media sources or making decisions by combining data from various sources. For instance, in the field of autonomous driving technology, autonomous vehicles gather data from different perspectives through sensors such as cameras and radars, integrating and analyzing these data to make intelligent decisions. These diverse and rich sources of information present us with the diversity and complementary views of the data. Therefore, effectively integrating and mining the rich information contained in these multimedia data sources becomes a crucial issue.

*ACM MM, 2024, Melbourne, Australia*
© 2024 Copyright held by the owner/author(s). Publication rights licensed to ACM.
ACM ISBN 978-x-xxxx-xxxx-x/YY/MM
https://doi.org/10.1145/nnnnnnn.nnnnnnn

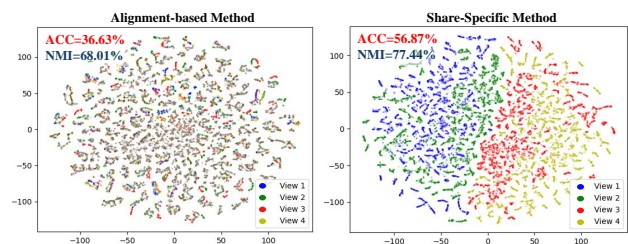

(a) t-SNE of alignment-based method (b) t-SNE of share-specific method

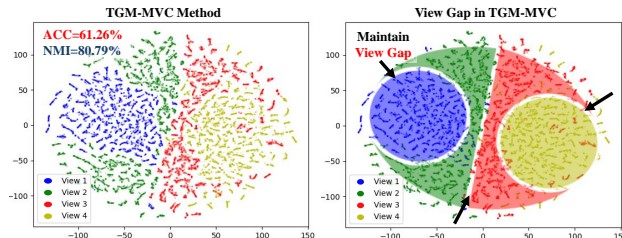

(c) t-SNE of TGM-MVC method (d) view gap illustration on TGM-MVC

**Figure 1: t-SNE Visualization on ALOI-100 dataset, where each color represents a view. Fig (a) presents the alignment-based method. Fig (b) presents the share-specific method. Fig (c) presents our proposed TGM-MVC method. Fig (d) demonstrates the view gap maintained by the TGM-MVC method. By training these three methods for same epochs, we have verified that maintaining the view gap helps preserve view diversity and enhances clustering effectiveness.**

Multi-view learning is a nontrivial topic in the filed of multimedia technology. A fundamental issue of multi-view learning is multi-view representation learning [6, 12, 15, 17, 28, 32, 44], which addresses the challenge of unifying data representation from diverse sources. Multi-view clustering (MVC) [3, 16, 18, 40, 45, 48, 50, 52] is a typical task in multi-view learning, where the clustering performance is largely contingent upon the quality of the representations of multi-view samples. As such, exploring representation learning across multiple views holds significant importance, especially within the domain of multi-view clustering tasks.

Existing deep multi-view clustering representation learning methods can be categorized into three paradigms [9, 15], *i.e.*, joint methods [13, 39, 41, 55], alignment-based methods [6, 12, 32, 52], and share-specific methods [10, 27, 29, 44]. Joint methods enable samples from different views to be independently optimized within their respective view spaces, and then integrates the representations from different views through fusion techniques. The latter two methods both map representations from different views into a shared subspace, aiming to learn the unified representation for all views. Alignment-based methods [6, 12, 32, 52] aim to bring representations closer between any view pair. Share-specific methods

[10, 27, 29, 44] model the view information as shared information and specific information, maximizing the mutual information of shared information across any view.

Alignment-based methods and share-specific methods have demonstrated notable advancements in the MVC task. However, their capacity to fully apprehend the intricacies of view heterogeneity appears to be constrained. Alignment-based methods concentrate solely on the consensus between views, overlooking the complementary information among them. Share-specific methods forcibly bring views closer in consensus, and lack supervision in learning complementary information as well. In real-world scenarios, views exhibit intricate adjacency relationships. Fig. 1 presents t-SNE visualizations of different views from ALOI-100 dataset. In Fig. 1(a)(b)(c), we utilized alignment-based method, share-specific method, and our proposed TGM-MVC method for the same training epochs. The contrastive-based method homogenizes different views, leading to subpar clustering performance; whereas the TGM-MVC method demonstrates excellent clustering performance, maintaining the distinctions between views (Fig. 1(d)). In this paper, we define the views' distinctions as **view gap**, which is measured by cosine distance of view representations. Based on the concept of view gap, we propose a foundational hypothesis, *i.e.* the pursuit of uniform consensus across all views may be inherently irrational. Instead, **preserving the view gaps between distinct views and enriching their diversity should facility the learning of more robust multi-view representations**.

In line with the limitations highlighted above for current share-specific methods, we introduce a novel approach, termed the Tree-Based View Coordination Enhancement (TGM). Building upon the share-specific framework, our proposed method incorporates a more comprehensive understanding of the relationships between multi-views. Our core idea is the recognition of the intricate adjacency relationships that exist between views in real-world scenarios, as demonstrated by the t-SNE visualizations in Fig. 1(a) and (b). Unlike alignment-based methods that focus merely on achieving consensus or share-specific methods that force views into closer alignment, our TGM method takes into account both consensus and complementary information. To achieve this, we begin by constructing a view adjacency matrix for the shared representations, capturing the inherent relationships across different views. With this matrix, we generate a minimum spanning tree that encapsulates the shared information while preserving the natural distances between views. By bringing closer the shared representations between adjacent nodes in this view tree, we avoid the limitations of directly aligning distant views, as observed in Fig. 1(d).

The primary contributions of this work can be summarized as follows:

- Our TGM-MVC method acknowledges the intrinsic "view gap" between views. Through theoretical and empirical research, we demonstrate that maintaining this view gap contributes to preserving the rich information of multi-views.
- The tree-based view graph generation strategy allows for the rapid and efficient construction of neighborhood relationships among views, analyzing the proximity relationships between views, thus facilitating downstream tasks in multi-view learning.

- Comprehensive experiments across six benchmark datasets serves to highlight both the superiority and efficiency of the proposed TGM-MVC method. Furthermore, the effectiveness of our approach is substantiated by ablation studies and visualization experiments.

## 2 RELATED WORK

### 2.1 Deep Multi-view Representation Learning

Recently, Multi-View Clustering (MVC) [1, 20–26, 34, 37, 38, 51] has recently attracted considerable attention as a crucial multimedia technology. Within this domain, deep Multi-View Clustering (DMVC) [2, 8, 18, 19, 30, 32, 42, 46, 53] using deep networks has emerged as an important approach. Presently, deep multi-view clustering methods can be categorized into three main classes: joint methods, alignment-based methods, and share-specific methods. Joint methods consider the differences and complementarities between views, achieving the representations by independently optimizing and concatenating sample representations in each view space. For instance, DMJC [41] involves training independent autoencoders for each view and then utilizes sharpening of the distribution of concatenated representations as a self-supervised signal for training. While joint method for multi-view representation learning is straightforward and effective, it lacks direct interaction between views, thereby hindering the acquisition of a consensus across multiple views. Alignment-based methods, on the other hand, map representations from different views to a shared semantic space based on the consistency of multi-view data, with the most typical approach being contrastive learning to minimize the distances between any two views. For instance, MFLVC [47] introduces two objectives on high-level features and pseudo-labels, leveraging contrastive learning to diminish the distances between views and achieve multi-view clustering. However, this method primarily focuses on the consensus among views, neglecting the differences and gaps between views, leading to the loss of unique view-specific information during the alignment process and resulting in information loss. Share-specific methods offer a more comprehensive integration of the aforementioned approaches by decoupling representations into consensus information and unique information, thus considering both the consistency and complementarity of multiple views. Nevertheless, the share-specific information architecture still exhibits shortcomings in addressing the view gap. In the subsequent section, we will delve into a detailed illustration of the view gap issue.

### 2.2 Rethinking of view gap in DMVC

Recent studies have unveiled the existence of gaps among heterogeneous data sources, and forcefully eradicating these distinctions could detrimentally affect the data's representation learning. Wang et al. [36] carried out a theoretical scrutiny on contrastive loss, stressing that enhanced alignment should involve the diminishment of disparities across diverse modalities. Nonetheless, although alignment is extensively applied in pre-training utilizing multiple data sources, potential conflicts could emerge between upstream alignment objectives and diverse downstream tasks, such as classification or clustering. Jiang et al. [11] analyzed two modalities, image and text, and discovered that minimizing the modality gaps

**Table 1: Basic notations used in this paper.**

| Notation | Meaning |
|---|---|
| $\mathbf{X}^{(v)}$ | Data matrix of the $v$-th view |
| $\widetilde{\mathbf{Z}}^{(v)}$ | Shared representation of the $v$-th view |
| $\hat{\mathbf{Z}}^{(v)}$ | Specific representation of the $v$-th view |
| $\mathbf{Z}$ | Global representation of all views |
| $V$ | View number of the multi-view data set |
| $N$ | Sample number of the multi-view data set |
| $\widetilde{E}, \widetilde{D}$ | Shared Encoder/Decoder for all views |
| $\hat{E}^{(v)}, \hat{D}^{(v)}$ | Specific Encoder/Decoder for the $v$-th view |
| $\widetilde{\mathbf{M}}$ | View Consensus Distance Matrix |
| $\hat{\mathbf{M}}$ | View Heterogeneity Distance Matrix |
| $\mathcal{G} = (\mathcal{X}, \mathcal{E})$ | Fully-connected view graph |
| $\widetilde{\mathcal{G}} = (\mathcal{X}, \widetilde{\mathcal{E}})$ | View Consensus Spanning Tree |
| $\hat{\mathcal{G}} = (\mathcal{X}, \hat{\mathcal{E}})$ | View Heterogeneity Spanning Tree |

does not always lead to improved performance in subsequent tasks. While existing studies have primarily concentrated on dual data sources, when dealing with multiple data sources, as in multi-view settings, the variations in distances between different view pairs can pose a growing challenge in distance evaluation. Dong et al. [4] constructed a relational matrix among views rooted in the distribution of view data, with the intention of fostering consensus among views while simultaneously deviating from specialized complementary representations linked to specific views as indicated by the relationship matrix. Nevertheless, the diverse gaps among different view pairs suggest an inherent complexity in both harmonizing all views towards consensus and distancing complementary representations within diverse views. Tackling the intricacies stemming from the disparities among various views in a multi-view context continues to pose an unresolved challenge.

## 3 METHOD

In this section, we elaborate on our Tree-Based View-Gap Maintaining Multi-View Clustering (TGM-MVC) method. The entire framework is illustrated in Fig. 2. It encompasses three modules and three stages: 1) In the first stage, we employ a conventional share-specific Learning Module (**ShaSpec**) to decouple sample features $\mathbf{X}^{(v)}$ into shared representations $\widetilde{\mathbf{Z}}^{(v)}$ and specific representations $\hat{\mathbf{Z}}^{(v)}$. 2) In the second stage, we introduce a Shared Tree-based View Consensus Learning Module (**ShaTree**) and a Specific Tree-based View Gap Maintaining Module (**SpecTree**), which respectively operate on the shared representations $\widetilde{\mathbf{Z}}^{(v)}$ and specific representations $\hat{\mathbf{Z}}^{(v)}$ obtained from the first module. 3) In the third stage, we concatenate shared and specific representations from all views for clustering. For clarity, all symbols and their meanings are presented in Table 1.

### 3.1 Problem Formulation

Given a set of multiview data $\mathcal{X} = \{\mathbf{X}^{(1)}, \mathbf{X}^{(2)}, ..., \mathbf{X}^{(V)}\}$, where $V$ is the number of views. $\mathbf{X}^{(v)} = \{\mathbf{x}_1^{(v)}, \mathbf{x}_2^{(v)}, ..., \mathbf{x}_N^{(v)}\} \in \mathbb{R}^{N \times d_v}$ represents the sample set of the $v$-th view, with $N$ as the number of samples, and $d_v$ as the input dimension of the $v$-th view. $\mathbf{x}_i^{(v)}$ denotes the $i$-th sample of the $v$-th view.

We extract sample $\mathbf{x}_i^{(v)}$ into two types of features, that is the shared view feature $\widetilde{\mathbf{z}}_i^{(v)}$ and the specific view feature $\hat{\mathbf{z}}_i^{(v)}$, in the manner of [10, 27, 29, 44]. To achieve this, we trained specific encoders $\hat{\mathbf{E}}^{(v)}$ and specific decoders $\hat{\mathbf{D}}^{(v)}$ tailored to each view, along with a universal shared encoder $\widetilde{\mathbf{E}}$ and a shared decoder $\widetilde{\mathbf{D}}$ for all views. It should be noted that in $\widetilde{\mathbf{E}}$ or $\widetilde{\mathbf{D}}$, all components are the same across all views except for the first and last two linear mapping layers. The ultimate objective is to achieve the optimal clustering performance by concatenating the shared and specific representations from individual views.

### 3.2 Share-Specific Representation Learning Module

To exploit the consensus and heterogeneity between views, we leverage the share-specific representation learning module (**ShaSpec**) in the Stage 1 to decouple the sample features into shared representations and specific representations. The loss within the ShaSpec module is composed of three components: reconstruction loss $L_{rec}$, contrastive learning loss $L_{con}$, and orthogonal loss $L_{oth}$. We begin by decoupling the representations of the samples, with

$$\widetilde{\mathbf{Z}}^{(v)} = \widetilde{E}\left(\mathbf{X}^{(v)}\right), \quad and \quad \hat{\mathbf{Z}}^{(v)} = \hat{E}^{(v)}\left(\mathbf{X}^{(v)}\right), \quad (1)$$

for $i \in \{1, ..., N\}$ and $v \in \{1, ..., V\}$. To prevent feature collapse, we trained a corresponding decoder for each encoder to reconstruct the sample features, with

$$\widetilde{\mathbf{X}}^{(v)} = \widetilde{D}\left(\widetilde{\mathbf{Z}}^{(v)}\right), \quad and \quad \hat{\mathbf{X}}^{(v)} = \hat{D}^{(v)}\left(\hat{\mathbf{Z}}^{(v)}\right), \quad (2)$$

where $\widetilde{\mathbf{X}}^{(v)}$ and $\hat{\mathbf{X}}^{(v)}$ respectively represent the reconstruction of shared representation $\widetilde{\mathbf{Z}}^{(v)}$ and specific representation $\hat{\mathbf{Z}}^{(v)}$ on the $v$-th view. Then we optimize the reconstruction loss of all views by

$$\mathcal{L}_{rec} = \sum_{v=1}^{V} \sum_{i=1}^{N} (\| \widetilde{\mathbf{x}}_i^{(v)} - \mathbf{x}_i^{(v)} \|_2^2 + \| \hat{\mathbf{x}}_i^{(v)} - \mathbf{x}_i^{(v)} \|_2^2), \quad (3)$$

where the reconstruction loss of shared and specific representations from all views are aggregated.

**Shared Representations Learning:** The **ShaSpec** module focuses on the learning of shared representations $\{\widetilde{\mathbf{Z}}^{(v)}\}_{v=1}^{V}$ using contrastive learning to gather consensus information among views. Its objective is to minimize the differences between shared representations from different views, thereby bringing them closer together. The contrastive loss $\ell_{ins}(p, q)$ between the $p$-th view and the $q$-th view is represented as:

$$\ell_i^{(p)} = -\log \frac{\exp(\int(\widetilde{\mathbf{z}}_i^{(p)}, \widetilde{\mathbf{z}}_i^{(q)})/\tau_l)}{\sum_{j=1}^{N} \left[\exp(\int(\widetilde{\mathbf{z}}_i^{(p)}, \widetilde{\mathbf{z}}_j^{(p)})/\tau_l) + \exp(\int(\widetilde{\mathbf{z}}_i^{(p)}, \widetilde{\mathbf{z}}_j^{(q)})/\tau_l)\right]}, \quad (4)$$

$$\ell_{ins}(p, q) = \frac{1}{2N} \sum_{i=1}^{N} \left(\ell_i^{(p)} + \ell_i^{(q)}\right), \quad (5)$$

where $\int(\cdot, \cdot)$ is the cosine similarity and $\tau_l$ is the temperature hypothesis. Then the overall contrastive loss $\mathcal{L}_{con}$ between any two views is given by:

$$\mathcal{L}_{con} = \sum_{p=1}^{V-1} \sum_{q=p+1}^{V} \ell_{ins}(p, q). \quad (6)$$

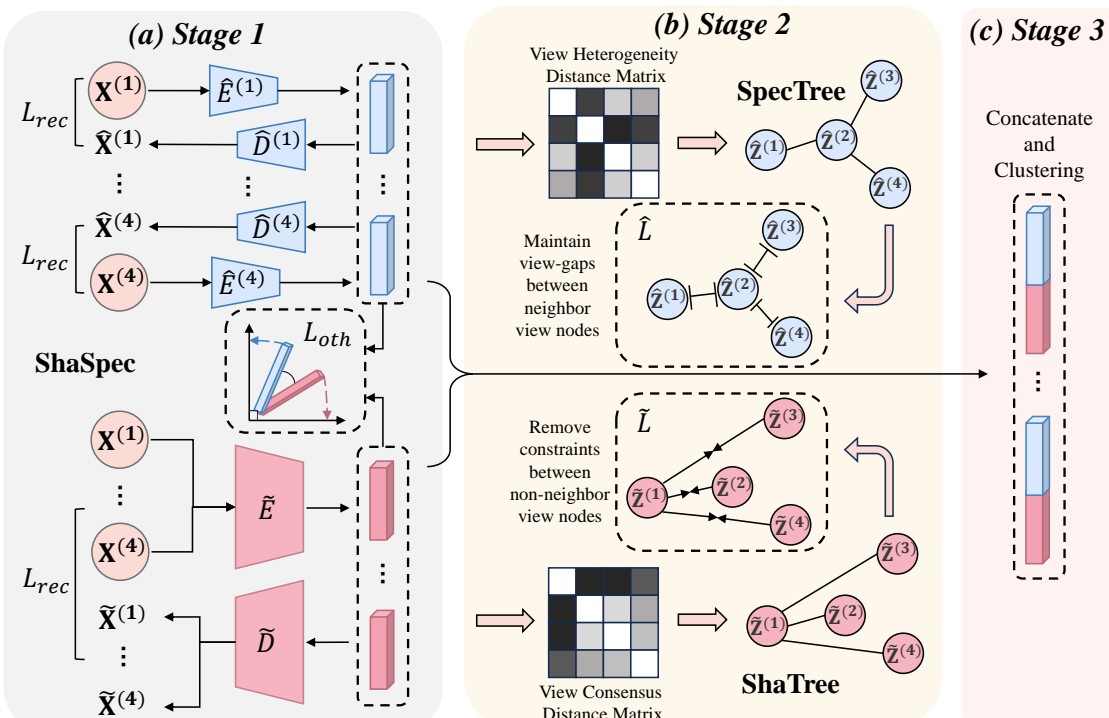

**Figure 2: Illustration of the Tree-Based View-Gap Maintaining Multi-View Clustering(TGM-MVC) method. Our framework consists of 3 stages with 3 modules. In Stage 1, we employ the shared-specific learning module (ShaSpec) to disentangle the representation of samples on each view into shared representations and view-specific representations. In Stage 2, we utilize the Shared Tree-based View Consensus Learning Module (ShaTree) to create the View Consensus Distance Matrix using the shared representations derived in Stage 1. Subsequently, a View Consensus Spanning Tree$\widetilde{\mathcal{G}}$ is established, narrowing down the shared representations of adjacent view pairs on the tree. Similarly, the Specific Tree-based View Gap Maintaining Module (SpecTree) constructs a View Heterogeneity Spanning Tree based on specific representations to preserve the gaps between adjacent views. The shared and specific representations from all views are concatenated for clustering at Stage 3.**

**Specific Representations Learning:** For specific representations $\{\hat{\mathbf{Z}}^{(v)}\}_{v=1}^{V}$, **ShaSpec** module aims to acquire view heterogeneity by making the $\hat{\mathbf{Z}}^{(v)}$ orthogonal to the shared representation $\widetilde{\mathbf{Z}}^{(v)}$ of that view, thereby designing the loss for specific representations as:

$$\mathcal{L}_{oth} = \sum_{v=1}^{V} \sum_{i=1}^{N} \int(\widetilde{\mathbf{z}}_i^{(v)}, \hat{\mathbf{z}}_i^{(v)}), \tag{7}$$

where $\int(\cdot, \cdot)$ represents the cosine similarity between sample features.

In Stage 1, we train the **ShaSpec** module for $T_1$ rounds to decouple samples into view consensus information and view heterogeneity information, enabling a detailed study of the relationships between views in the subsequent stage. The overall loss of Stage 1 learning consists of the reconstruction loss $\mathcal{L}_{rec}$, the contrastive loss $\mathcal{L}_{con}$ and the orthogonal loss $\mathcal{L}_{oth}$, i.e.,

$$\mathcal{L}_{pre} = \mathcal{L}_{rec} + \lambda_1 \cdot \mathcal{L}_{oth} + \lambda_2 \cdot \mathcal{L}_{con}, \tag{8}$$

where $\mathcal{L}_{oth}$ and $\mathcal{L}_{con}$ with $\lambda_1$ and $\lambda_2$ are the weights balancing the three loss terms.

### 3.3 Shared Tree-based View Consensus Learning Module

During the Stage 1, we aim to align the consensus representations of all views using a contrastive learning paradigm to achieve uniform representations across all views. However, based on our assumption, there exists view gaps between different views. Disregarding the view gap and forcibly aligning the representations of arbitrary pairs of views would lead to the loss of rich multi-view information. Therefore, we seek to rectify the consensus representations obtained from Stage 1 through the Shared Tree-based View Consensus Learning Module (**ShaTree**), obtaining consensus from adjacent view pairs instead of all views.

To quantify the proximity relationships between views, we construct the View Consensus Distance Matrix $\widetilde{\mathbf{M}} \in \mathbb{R}^{V \times V}$, where $\widetilde{\mathbf{M}}_{pq}$ represents the cosine similarity distance of the shared representations between the $p$-th view and the $q$-th view:

$$\widetilde{\mathbf{M}}_{pq} = \frac{1}{N} \sum_{i=1}^{N} (1 - \int(\widetilde{\mathbf{z}}_i^{(p)}, \widetilde{\mathbf{z}}_i^{(q)})), \tag{9}$$

where the distance between views is defined as the average cosine similarity distance between corresponding samples of the two views.

If we we conceptualize multiple views as a graph structure, considering each view as a node and the distance between views as the edge length between nodes, then the multi-view can be modeled as a view graph $\mathcal{G} = (\mathcal{X}, \mathcal{E})$, which contains $V$ nodes $\mathcal{X} = \{\mathbf{X}^{(1)}, \mathbf{X}^{(2)}, ..., \mathbf{X}^{(V)}\}$ and $C_V^2$ edges $\mathcal{E} = \{(\mathbf{X}^{(p)}, \mathbf{X}^{(q)}) \mid p, q \in [1, V], p < q\}$. To simplify the representation, we denote the edge $(\mathbf{X}^{(p)}, \mathbf{X}^{(q)})$ as $(p, q)$.

To discover the neighborhood relationships between consensus information of views, we employ the Prim algorithm [5, 7] on the View Consensus Distance Matrix $\widetilde{\mathbf{M}}$ to generate a minimum spanning tree, obtaining a subgraph $\widetilde{\mathcal{G}} = (\mathcal{X}, \widetilde{\mathcal{E}})$ of $\mathcal{G}$, which edge set $\widetilde{\mathcal{E}} = \{(p_1, q_1), (p_2, q_2), ..., (p_{V-1}, q_{V-1}))\}$.

To guarantee diversity in view information and avoid the direct merging of shared representations from distant views, we exclusively bring closer the shared representations among view nodes that correspond to the edges encompassed in the View Consensus Spanning Tree $\widetilde{\mathcal{G}}$. Hence, we adjust Eq.(6) in the Stage 1 as:

$$\widetilde{\mathcal{L}} = \sum_{(p,q) \in \widetilde{\mathcal{E}}} \ell_{ins}(p, q), \tag{10}$$

where $\ell_{ins}(p, q)$ is the contrastive loss between two views, *i.e.*, $\mathbf{X}^{(p)}$ and $\mathbf{X}^{(q)}$, as computed in Eq.(5).

## 3.4 Specific Tree-based View Gap Maintaining Module

The specific representation $\hat{\mathbf{Z}}^{(v)}$ learned in Stage 1 is primarily constrained by the orthogonal loss and the reconstruction loss. There remains a risk that the specific representations of various views lack the necessary distinctiveness since the above two constraints can hardly ensure that the learned $\hat{\mathbf{Z}}^{(v)}$ fully captures the heterogeneity inherent in different views. Given the insights from pertinent information theory [3], we introduce the concept of **information gap** $\delta_{pq}$ to highlight the potential consequences of homogenizing the representations across different views for downstream tasks. Such $\delta_{pq}$ indicates the difference in the amount of information provided by the two views for the clustering task. We introduce the theoretical optimal output $\mathbf{Y}^*$ in the solution space, such that the information gap $\delta_{pq}$ can be defined as:

$$\delta_{pq} = |I(\mathbf{X}^{(p)}; \mathbf{Y}^*) - I(\mathbf{X}^{(q)}; \mathbf{Y}^*)|, \tag{11}$$

where $\mathbf{Y}^* = \arg\max_{\mathbf{Y}^*} NMI(\mathbf{Y}^*, \mathbf{Y})$. $I(\mathbf{X}^{(v)}; \mathbf{Y}^*)$ illustrates the amount of information that the $v$-th view can contribute to the clustering task. It is noteworthy that $\mathbf{Y}$ is a clustering label used for theoretical elucidation, thus our task remains inherently unsupervised.

**Theorem 1.** Suppose there exists encoders $h_p : \mathbf{X}^{(p)} \to \mathbf{Z}^{(p)}$ and $h_q : \mathbf{X}^{(q)} \to \mathbf{Z}^{(q)}$, such that $\mathbf{Z}^{(p)} = \mathbf{Z}^{(q)}$. And the fusion functions $g_x$ and $g_z$, applying to $\{X^{(v)}\}_{v=p,q}$ and $\{Z^{(v)}\}_{v=p,q}$ respectively, allow the fused features to retain maximum information content. Then encoders $h_p$ and $h_q$ would disregard the view gap, resulting in information loss:

$$I(g_x(\mathbf{X}^{(p)}, \mathbf{X}^{(q)}); \mathbf{Y}^*) - I(g_z(\mathbf{Z}^{(p)}, \mathbf{Z}^{(q)}); \mathbf{Y}^*) \geq \delta_{pq}. \tag{12}$$

**Theorem 1** elucidates that as the representations of two views converge completely, it leads to information loss and impairs the performance of downstream tasks. Additional theoretical analysis can be found in Appendix A.

According to [36], contrastive learning continuously narrows the gap between the two views when a plentiful amount of negative samples are available. To preserve the view gap, apart from avoiding a direct convergence of shared representations $\widetilde{\mathbf{Z}}^{(v)}$ between distant views, we also impose constraints on the view-specific features using a similar approach. Specifically, we generate the View Heterogeneity Distance Matrix $\hat{\mathbf{M}} \in \mathbb{R}^{V \times V}$ based on the view-specific representations $\hat{\mathbf{Z}}^{(v)}$ obtained in the Stage 1:

$$\hat{\mathbf{M}}_{pq} = \frac{1}{N} \sum_{i=1}^{N} (1 - \int(\hat{\mathbf{z}}_i^{(p)}, \hat{\mathbf{z}}_i^{(q)})), \tag{13}$$

where $\int(\hat{\mathbf{z}}_i^{(p)}, \hat{\mathbf{z}}_i^{(q)})$ is the cosine similarity between the two representations.

Similar to the View Consensus Spanning Tree $\widetilde{\mathcal{G}}$, we construct a View Heterogeneity Spanning Tree $\hat{\mathcal{G}} = (\mathcal{X}, \hat{\mathcal{E}})$ based on the View Heterogeneity Distance Matrix $\hat{\mathbf{M}}$ using the Prim algorithm, where the pair of views $(p, q)$ in $\hat{\mathcal{E}}$ indicates a higher similarity in specific representations for the $p$-th view and the $q$-th view. To maintain the view gap between these two views, we design the loss as:

$$\ell_i^{(p)} = -\log \frac{\exp(\int(\hat{\mathbf{z}}_i^{(p)}, \hat{\mathbf{z}}_i^{(q)})/\tau_l)}{\sum_{j=1}^{N} \left[\exp(\int(\hat{\mathbf{z}}_i^{(p)}, \hat{\mathbf{z}}_j^{(p)})/\tau_l) + \exp(\int(\hat{\mathbf{z}}_i^{(p)}, \hat{\mathbf{z}}_j^{(q)})/\tau_l)\right]}, \tag{14}$$

$$\ell_{sp}(p, q) = \sqrt{\frac{1}{2N} \sum_{v=p,q} \sum_{i=1}^{N} \left(\hat{\ell}_i^{(v)} - \hat{\ell}_\mu\right)^2}, \tag{15}$$

where $\hat{\ell}_\mu = \frac{1}{2N} \sum_{i=1}^{N} \left(\hat{\ell}_i^{(p)} + \hat{\ell}_i^{(q)}\right)$. Hence, the purpose of $\ell_{sp}(p, q)$ is to minimize the variance of the contrastive losses incurred by any sample in the $p$-th view and the $q$-th view.

**Theorem 2.** Imposing the constraints of $\ell_{sp}(p, q)$ can prevent the convergence of representations from two views into uniformity, i.e. $\hat{\mathbf{Z}}^{(p)} \neq \hat{\mathbf{Z}}^{(q)}$, thereby preserving the diverse information present across the multiple views.

The proof of **Theorem 2** is relocated to the Appendix B. In regard to view pairs within View Heterogeneity Spanning Tree $\hat{\mathcal{G}}$, we impose constraints using Eq.(15), thereby crafting the loss for specific representations as:

$$\hat{\mathcal{L}} = \sum_{(p,q) \in \hat{\mathcal{E}}} \ell_{sp}(p, q). \tag{16}$$

## 3.5 Implementation

In Stage 1, we solely employ the **ShaSpec** module for training, decoupling the representations into view-shared and view-specific representations using Eq.(8).

**Algorithm 1** Tree-Based View-Gap Maintaining Multi-View Clustering(TGM-MVC)

---

**Input**: The multi-view raw features $\{\mathbf{X}^{(v)}\}_{v=1}^{V}$; the interation number $T_1$ and $T_2$.

**Output**: The clustering result $\mathbf{R}$.

1: **for** $i = 1$ to $T_1$ **do**
2:     Obtain the shared representations $\{\widetilde{\mathbf{Z}}^{(v)}\}_{v=1}^{V}$ and the specific representations $\{\hat{\mathbf{Z}}^{(v)}\}_{v=1}^{V}$ by **ShaSpec** Module using Eq.(8).
3: **end for**
4: **while** the total loss hasn't converged **do**
5:     Calculate View Consensus Distance Matrix $\widetilde{\mathbf{M}}$ and View Heterogeneity Distance Matrix $\hat{\mathbf{M}}$ using Eq.(9)(13).
6:     Obtain View Consensus Spanning Tree $\widetilde{\mathcal{G}} = (\mathcal{X}, \widetilde{\mathcal{E}})$ and View Heterogeneity Spanning Tree $\hat{\mathcal{G}} = (\mathcal{X}, \hat{\mathcal{E}})$ using the Prim algorithm.
7:     **for** $j = 1$ to $T_2$ **do**
8:         Calculate the total loss $L_{total}$ using Eq.(17).
9:         Updating the network with Adam Optimizer by minimizing $L_{total}$.
10:     **end for**
11: **end while**
12: Concatenate shared and specific representations across all views using Eq.(18) and then clustering.
13: **return** $\mathbf{R}$

---

**Table 2: Statistics summary of eight datasets.**

| Dataset | Samples | Clusters | Views |
|---|---|---|---|
| Synthetic3d | 600 | 3 | 3 |
| Cora | 2708 | 7 | 4 |
| ReutersEN | 7200 | 6 | 5 |
| Caltech101 | 9144 | 102 | 5 |
| ALOI-100 | 10800 | 100 | 4 |
| STL10 | 13000 | 10 | 4 |

To uphold the view gaps between views and ensure the richness of information across multiple views, we calibrate Stage 1 through Stage 2. Every $T_2$ rounds, we reassess the View Consensus Spanning Tree and View Heterogeneity Spanning Tree, optimizing the entirety through reconstruction loss, orthogonality loss, and two-part losses acting upon shared and specific representations:

$$\mathcal{L}_{total} = \mathcal{L}_{rec} + \lambda_1 \cdot \mathcal{L}_{oth} + \lambda_2 \cdot \widetilde{\mathcal{L}} + \lambda_3 \cdot \hat{\mathcal{L}}, \quad (17)$$

where $\lambda_1$, $\lambda_2$ and $\lambda_3$ are the weights for different parts of loss respectively. Meanwhile, the contrastive loss $\mathcal{L}_{con}$ in Stage 1 is replaced by $\widetilde{\mathcal{L}}$ in Stage 2.

Upon completion of training in Stage 2, we concatenate the shared and specific representations across all views to derive the holistic representation $\mathbf{Z}$ of the samples, which is subsequently employed for the final clustering task:

$$\mathbf{Z} = \left(\bigcup_{v=1}^{V} \widetilde{\mathbf{Z}}^{(v)}\right) \oplus \left(\bigcup_{v=1}^{V} \hat{\mathbf{Z}}^{(v)}\right), \quad (18)$$

where symbol $\bigcup$ and $\oplus$ both denote the concatenation on the feature dimension. The detailed learning process of our proposed TGM-MVC is shown in **Algorithm 1**.

## 4 EXPERIMENTS

### 4.1 Datasets

To demonstrate the efficacy of our TGM-MVC approach, we conduct elaborate experiments on six benchmark datasets: Synthetic3d, Cora, ReutersEN, Caltech101, ALOI-100, and STL10. The fundamental characteristics of these six datasets are illustrated in Table 2. In this section, we validate the effectiveness of our proposed TGM-MVC by addressing the following five inquiries:

- **Q1**: How effective does the TGM-MVC method exhibit in the realm of deep multi-view clustering tasks?
- **Q2**: How does the View Consensus Spanning Tree influence the performance of TGM-MVC?
- **Q3**: How does the View Heterogeneity Spanning Tree influence the performance of TGM-MVC?
- **Q4**: How do the hyper-parameters impact the performance of TGM-MVC?
- **Q5**: Does the TGM-MVC model exhibit convergence?

### 4.2 Experiment Settings

The experimental setup includes an Intel Core i7-7820x CPU, NVIDIA GeForce RTX 3090 GPU, and 64GB of RAM. For software support, the experiments were carried out using the PyCharm platform. Additionally, the training process made use of the Adam optimizer.

*4.2.1 Compared Methods.* We conducted a comparative analysis between the TGM-MVC method and eight state-of-the-art multi-view clustering algorithms: DEMVC [43], FMCNOF [49], CoMVC [32], SiMVC [32], SDMVC [45], MFLVC [47], DSMVC [31], and SFMC [14].

*4.2.2 Evaluation metrics.* In order to assess the efficacy and superiority of our TGM-MVC approach, we utilize commonly employed metrics, namely clustering accuracy (ACC), normalized mutual information (NMI), and purity (PUR) [33, 35, 54].

### 4.3 Performance Comparison (Q1)

To validate the efficacy of our proposed method, we conducted comparisons with eight benchmark methods across six datasets. The clustering results of our model and other algorithms are presented in Table 3. From Table 3, the following conclusions can be drawn:

(1) Our approach outperformed or matched existing state-of-the-art methods in three metrics across the six datasets. With respect to Purity, our method outshines the baseline algorithms, especially on the Caltech101 and ALOI-100 dataset, which surpasses the second-best algorithm by 8.55% and 6.22%.

(2) These datasets cover multiple views (ranging from 3 to 5 views), and our method excelled on most of these datasets, whereas other methods achieved satisfactory clustering results on only one or some of the datasets. We attribute this to our methodology's acknowledgment of the view gaps, thereby demonstrating strong performance on datasets with more views.

### 4.4 Ablation Studies (Q2 & Q3)

To verify the effectiveness of the **ShaTree** module and **SpecTree** module, we conducted ablation experiments on six datasets. Specifically, we held constant the Share-Specific Representation Learning

**Table 3: Clustering performance across six multi-view benchmark datasets. The most outstanding results are denoted in bold, while the second-best values are underlined. '-' indicates the error of the method itself.**

| Datasets | Methods | | | | | | | | |
|---|---|---|---|---|---|---|---|---|---|
| | DEMVC | FMCNOF | CoMVC | SiMVC | SDMVC | MFLVC | DSMVC | SFMC | **TGM-MVC** |
| ACC(%) | | | | | | | | | |
| Synthetic3d | 71.50 | 58.33 | 38.83 | 48.00 | 97.33 | 97.67 | 77.00 | 93.17 | **97.83** |
| Cora | 30.54 | 30.28 | 29.76 | 23.08 | 31.06 | 31.02 | 28.88 | 30.50 | **37.59** |
| ReutersEn | 34.58 | 19.25 | 21.11 | 20.25 | 29.32 | 25.42 | 32.04 | 16.00 | **34.78** |
| Caltech101 | 11.05 | 12.77 | 16.36 | 13.48 | 15.11 | **21.30** | 16.27 | 15.38 | 20.31 |
| ALOI-100 | - | 6.870 | 16.63 | 11.69 | - | 23.12 | 14.89 | 57.76 | **61.26** |
| STL10 | 30.01 | 25.09 | 23.55 | 16.04 | 28.34 | 31.14 | 27.53 | 10.06 | **32.13** |
| NMI(%) | | | | | | | | | |
| Synthetic3d | 54.81 | 13.59 | 6.920 | 23.47 | 88.19 | 89.64 | 44.53 | 76.96 | **90.51** |
| Cora | 6.340 | 7.270 | 4.640 | 3.010 | 5.120 | 12.97 | 8.140 | 7.95 | **13.93** |
| ReutersEn | 11.84 | 0.640 | 1.720 | 0.790 | 7.630 | 3.250 | 9.360 | **12.86** | 12.34 |
| Caltech101 | 22.84 | 15.84 | 25.61 | 18.18 | 30.48 | 28.60 | 26.53 | 19.06 | **40.62** |
| ALOI-100 | - | 31.66 | 44.16 | 36.21 | - | 67.88 | 40.57 | 71.62 | **80.79** |
| STL10 | 25.27 | 20.95 | 16.26 | 6.340 | **26.19** | 25.36 | 19.33 | 0.200 | 25.93 |
| Purity(%) | | | | | | | | | |
| Synthetic3d | 71.50 | 58.33 | 38.83 | 49.67 | 97.33 | 97.67 | 77.00 | 93.17 | **97.83** |
| Cora | 34.56 | 32.57 | 33.94 | 30.61 | 32.05 | 38.22 | 36.04 | 33.27 | **38.44** |
| ReutersEn | 34.69 | 19.78 | 21.96 | 20.53 | 30.01 | 26.64 | 33.32 | 25.31 | **36.36** |
| Caltech101 | 19.66 | 16.94 | 23.71 | 17.91 | 28.78 | 28.23 | 25.60 | 19.27 | **37.33** |
| ALOI-100 | - | 6.960 | 16.65 | 11.84 | - | 23.12 | 15.79 | 60.00 | **66.22** |
| STL10 | 31.07 | 26.02 | 25.01 | 17.02 | 30.13 | 31.25 | 29.41 | 10.10 | **34.48** |

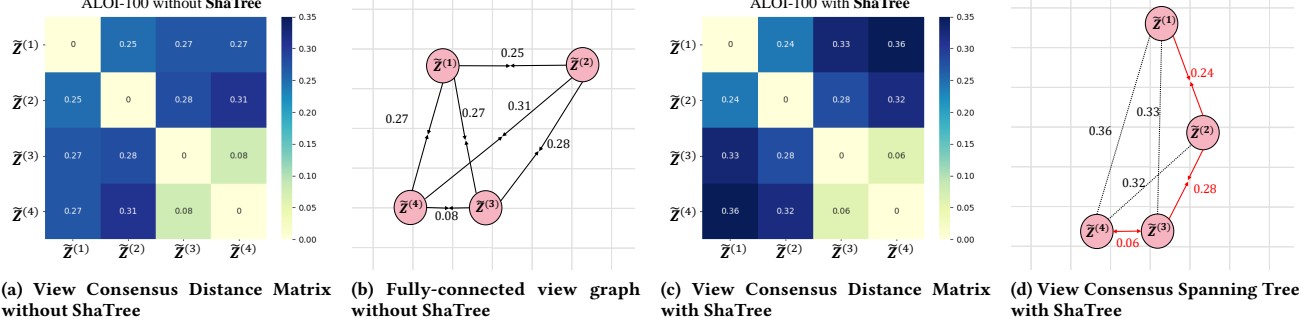

(a) View Consensus Distance Matrix without ShaTree

(b) Fully-connected view graph without ShaTree

(c) View Consensus Distance Matrix with ShaTree

(d) View Consensus Spanning Tree with ShaTree

**Figure 3: Visualization of View Consensus Distance Matrix and corresponding view graph on ALOI-100 dataset.**

Module and tested four scenarios: with or without **ShaTree** module and with or without the **SpecTree** module. Due to spatial constraints, we only present the results of the ablation experiments on the Synthetic3d, Reuters-7200, and ALOI-100 datasets, as illustrated in Table 4. Fig. 3,4 demonstrates the notable changes of ALOI-100 datasets between models with and without the two modules. The experimental results indicate the following conclusions:

(1) Solely employing the **ShaTree** module in Stage 2, without utilizing the **SpecTree** module, may lead to slight suboptimal outcomes for the model. This could be due to partial contrastive losses between view pairs being removed without additional constraints being introduced, leading to an overall lack of constraints on the model. As shown in Fig 3, the constraint between view 1 and view 4

has been released, leading to a notable increase in distance between the views.

(2) In Stage 2, while still aligning shared representations of arbitrary view pairs, the inclusion of **SpecTree** module effectively preserves the view gaps between views, enhancing clustering effects upon the **ShaSpec** framework. As depicted in Fig 4, utilizing the **SpecTree** module iteratively has significantly increased the distance between heterogeneous representations across views.

(3) The **ShaTree** module and **SpecTree** module synergize effectively, with the combined utilization of these two modules resulting in a greater improvement in clustering performance.

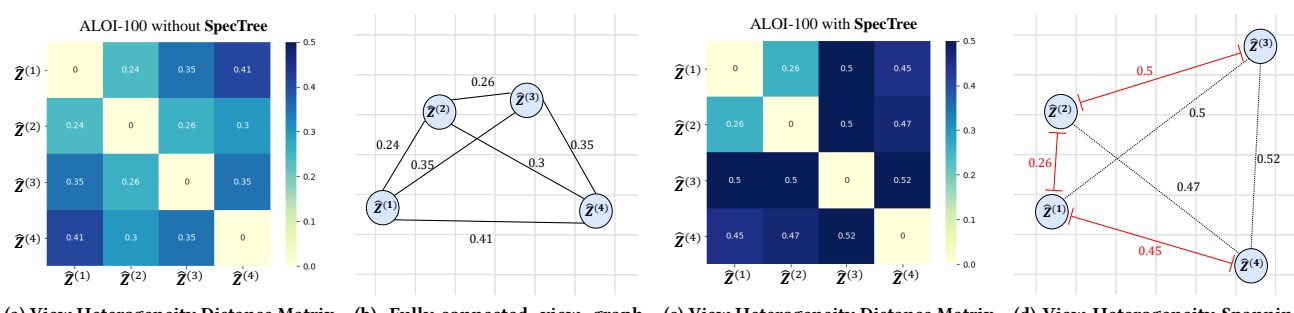

(a) View Heterogeneity Distance Matrix without SpecTree

(b) Fully-connected view graph without SpecTree

(c) View Heterogeneity Distance Matrix with SpecTree

(d) View Heterogeneity Spanning Tree with SpecTree

Figure 4: Visualization of View Heterogeneity Distance Matrix and corresponding view graph on ALOI-100 dataset.

Table 4: Ablation study on Synthetic3d, Reuters-7200 and ALOI-100 datasets. M1, M2 and M3 are abbreviations for ShaSpec Module(M1), ShaTree Module(M2) and SpecTree Module(M3), respectively. ✓ denotes TGM-MVC with the Module.

| Datasets | M1 | M2 | M3 | ACC | NMI | PUR |
|---|---|---|---|---|---|---|
| | ✓ | ✓ | ✓ | **97.83** | **90.51** | **97.83** |
| Synthetic3d | ✓ | | | 96.50 | 86.04 | 96.50 |
| | ✓ | | ✓ | 96.83 | 87.04 | 96.83 |
| | ✓ | ✓ | | 96.00 | 84.57 | 96.00 |
| | ✓ | ✓ | ✓ | **34.78** | **12.34** | **36.36** |
| Reuters-7200 | ✓ | | | 29.85 | 8.340 | 29.85 |
| | ✓ | | ✓ | 32.61 | 11.10 | 33.94 |
| | ✓ | ✓ | | 29.19 | 9.660 | 30.60 |
| | ✓ | ✓ | ✓ | **61.26** | **80.79** | **66.22** |
| ALOI-100 | ✓ | | | 56.87 | 77.44 | 62.52 |
| | ✓ | | ✓ | 58.66 | 79.77 | 64.72 |
| | ✓ | ✓ | | 53.94 | 76.09 | 59.95 |

## 4.5 Hyper-parameter Analysis (Q4)

In actual experiments, we kept the temperature hyperparameter $\tau_l$ for contrastive learning fixed at 0.1 and the weight $\lambda_1$ for orthogonal loss $\mathcal{L}_{oth}$ fixed at 1, adjusting only the weight $\lambda_2$ for shared representation loss $\widetilde{\mathcal{L}}$ and the weight $\lambda_3$ for specific representation loss $\hat{\mathcal{L}}$.

In order to assess the robustness of our model, we conducted a sensitivity analysis on $\lambda_2$ and $\lambda_3$. From the results in Fig. 5, we discern a notable influence of $\lambda_2$ on the model's performance, while the impact of $\lambda_3$ is relatively minor. This is because $\lambda_2$ is also involved in the training of the first stage, hence changes in $\lambda_2$ have a stronger impact on the overall training effectiveness of the model.

## 4.6 Analysis of Convergence Properties (Q5)

We completed experiments to demonstrate the convergence of the model on six datasets. Due to space constraints, we only demonstrate the changes in ACC and NMI with the number of training epochs on the ALOI-100 and Cora datasets, as shown in Fig 6. Experiments prove the convergence of our proposed TGM-MVC.

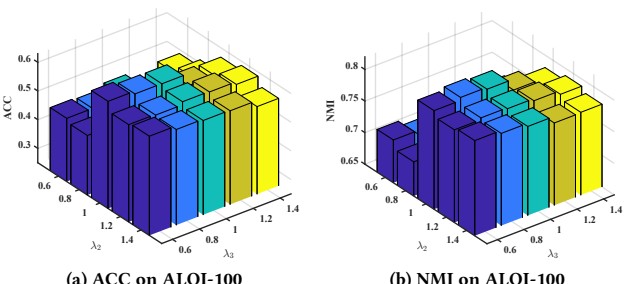

(a) ACC on ALOI-100

(b) NMI on ALOI-100

Figure 5: Sensitivity analysis of the hyper-parameters on ALOI-100 dataset.

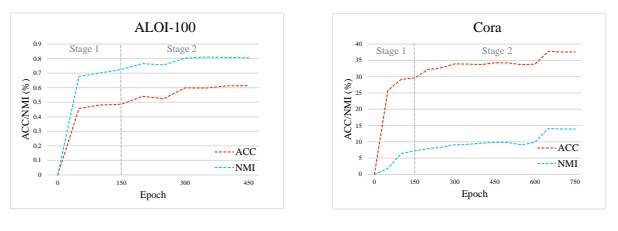

(a) Model Convergence on ALOI-100

(b) Model Convergence on Cora

Figure 6: ACC and NMI curves on the ALOI-100 and Cora.

## 5 CONCLUSIONS

In this paper, we introduce a novel approach for modeling multiple views utilizing a innovative spanning tree topology. Our proposed TGM-MVC method aims to preserve the distinctiveness among views, thereby enhancing the efficacy of multi-view clustering. Specifically, we separately generate minimum spanning trees for shared representations and specific representations, relaxing the constraints between non-adjacent views in shared representations while maintaining the distances between adjacent views in specific representations. We provide theoretical justification for the importance of maintaining view gaps and demonstrate the effectiveness of our approach through comprehensive experiments.

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
