# OpenReview forum: "View Gap Matters: Cross-view Topology and Information Decoupling for Multi-view Clustering"
_acmmm.org/ACMMM/2024/Conference — MM2024 Poster_

### Official Review · Reviewer_D1h8 · 2024-05-23

**Rating:** 6
**Confidence:** 4

**Summary:**

This manuscript proposes a Tree-Based View Gap Maintaining Multi-View Clustering (TGM-MVC) method. TGM-MVC reflects the affinity relationship between views by pruning the view graph, which provides a new insight for the MVC task. In addition, the sufficient experiments can verify the effectiveness of the approach.

**Strengths:**

Although there have been some MVC methods based on the decoupling of specific and common representations before, this manuscript is more suitable for processing the two representations based on the minimum spanning tree of the view. Therefore, I think this article is quite innovative. The experiments in this paper are comprehensive, which fully verify the effectiveness and superiority of the proposed method. This manuscript is well organized and clear for others to follow.

**Limitations:**

Regarding the understanding of view graph, I have a different opinion from the authors. First of all, I think that the view graph at the view level cannot measure the relationship between different views of all samples, because the proximity relationships between views are different at the fine-grained sample level. In addition, although the proposed method maintains the view gap, the most essential cluster structure of the cluster cannot be seen in the results in Figure 1.

In Stage 3 of Figure 2, it is depicted that all shared and specific representations are concatenated. However, the method used to derive the final clustering result from this amalgamation is not clearly explained. Please provide more details on how the final clustering outcome is obtained?

The placement of Q1 to Q5 proposed in section 4.1 is unreasonable, which should be placed between heading 4 and subheading 4.1.

The paper lacks an introduction to the comparative algorithms, among these algorithms I noticed that the SFMC algorithm exhibits significant performance differences across various datasets. Why does the performance of SOTA method SFMC drops dramatically on the STL 10 and ReutersEn datasets?

**Suitability:**

3

---

### Official Review · Reviewer_Rcsx · 2024-05-23

**Rating:** 5
**Confidence:** 4

**Summary:**

This paper proposes a Tree-Based Gap-Maintaining Multi-View Clustering (TGM-MVC) method. The approach introduces a novel concept of conceptualizing multiple views as a graph structure, where each view corresponds to a node and the differences between views as the edges. The minimum spanning tree is generated to reflect the neighboring relationships between views. Both theoretical analysis and comprehensive experiments demonstrate the effectiveness of our proposed method.

**Strengths:**

Novelty:I think that the perspective of maintaining the view gap between views presented in this paper is novel.

Theoretical Foundation:The theoretical foundation of the paper is robust, introducing the concept of the information gap to illustrate the problems with completely bringing different views closer together, thereby providing support for the perspective of maintaining the view gap.

Experiment Results:The experiments in the paper are thorough, utilizing topological graph visualizations of the view gap to demonstrate the effectiveness of the proposed ShaTree and SpecTree modules.

**Limitations:**

Visualization: In the visualization of the view gap, as depicted in Figure 1, it is intriguing to consider whether similar view gaps might be present in other datasets. To explore this further, it would be beneficial to conduct a comparative analysis using an additional dataset. Please provide another dataset to demonstrate whether this view gap phenomenon is consistent across different data contexts?

Experiments: In the ablation experiment (Section 4.4) results in Table 4, why are the results of using both ShaSpec and ShaTree modules worse than using only ShaSpec module? Does this mean that ShaTree has a negative effect on clustering? Please provide a detailed discussion of this phenomenon. The TCE-MVC method has three hyperparameters (lambda1, lambda2, lambda3). However, in the Hyper-parameter Analysis (Section 4.5), only lambda2 and lambda3 were analyzed. Why wasn't lambda1 analyzed? Please provide a sensitivity analysis for lambda1.

Citation:The six datasets used in the experiments lack citations, which I believe is not in accordance with standard practices.

**Suitability:**

3

---

### Official Review · Reviewer_vmE8 · 2024-05-24

**Rating:** 6
**Confidence:** 4

**Summary:**

This paper proposes a new method called TGM-MVC (Tree Based View Map Maintaining Multi View Clustering) for solving multi view clustering problems. Traditional multi view clustering methods typically attempt to reduce the distance between any two views, but overlook the heterogeneity between views, resulting in the loss of valuable information in multi view data. TGM-MVC constructs a graph structure between views, where each view is a node and the gaps between views (calculated by cosine distance) are used as edges. By pruning the graph, the minimum spanning tree of the view is obtained, reflecting the adjacency relationship between them.

**Strengths:**

The paper proposes a tree based view gap preserving multi view clustering (TGM-MVC) method, which views multiple views as graph structures, with each view corresponding to a node and the gaps between views as edges, allowing for modeling of heterogeneity between views. By pruning the graph, the minimum spanning tree of the view is obtained, reflecting the adjacency relationship between views, which helps to identify and preserve complementary information between different views.Adopting a shared specific learning framework, which only reduces the distance between adjacent views for shared information and maintains view gaps for specific information, is beneficial for maintaining view diversity.

**Limitations:**

The reference format is inconsistent, with some references having their year placed after the author's name, while others are placed at the end of the paper title.

**Suitability:**

3

---

### Official Review · Reviewer_JrNp · 2024-05-25

**Rating:** 5
**Confidence:** 4

**Summary:**

The author introduces the concept of view gap and acquires the adjacency relationships between views by generating a minimum spanning tree of the view graph. The shared and specific representations of samples are learned separately, ultimately achieving favorable clustering results.

**Strengths:**

1. The author proposes a novel definition of view graph. By considering views as nodes and the distances between views as edges, multi-view model is transformed into a graph structure. Then, by generating a minimum spanning tree, the adjacency relationships between views are represented using a tree structure.
2. The author proposes the perspective of maintaining the view gap, conducts theoretical analysis, and visualizes it in experiments. Additionally, through experimental results, the significance of maintaining the view gap for the Multi-View Clustering (MVC) task is demonstrated.
3. The proposed method works well on six datasets which contain different number of views, achieving significant performance gains compared to existing MVC competitors.

**Limitations:**

1. Learning the shared representations using the ShaSpec Module not only brings the neighboring views closer but also tends to bring the shared representations of all views closer together. Given this broader impact, why the approach isn't simply to bring together the shared representations across all views directly. Could you elaborate on the advantages of specifically employing the ShaSpec Module in this context?
2. MVC is inherently an unsupervised task, yet in the depiction of the information gap (Equation 11), Y* is derived from the label Y. Given this context, is it justifiable to use the label Y when defining the information gap?
3. The paper notes that the process of regenerating the minimum spanning tree within the model is iterative. Could you clarify how many iteration epochs are involved in this process?
4. The reference for the information gap (line 501) is "A novel approach for effective multi-view clustering with information-theoretic perspective.", which was made in error by mistake. The correct citation should be "Understanding and constructing latent modality structures in multi-modal representation learning."

**Suitability:**

2

---

### Official Review · Reviewer_u2iK · 2024-05-26

**Rating:** 4
**Confidence:** 4

**Summary:**

This paper proposes a novel multi-view clustering method called Tree-Based ViewGap Maintaining Multi-View Clustering (TGM-MVC). TGM-MVC considers multiple views as a graph structure, in which each view is a node, with the view gap is the edge. Then, the minimum spanning tree of the views can be derived by graph pruning, which can reflect the neighbouring relationships among views. For view-shared information, the distance between adjacent views should be narrowed. Meanwhile, for view-specific information, the distance between adjacent views should be maintained. Theoretical analysis and comprehensive experiments can show the effectiveness of the proposed TGM-MVC.

**Strengths:**

1.​The tree-based view learning is interesting and novelty, which can be readily implemented in deep multi-view learning framework.
2.​The framework is lucidly expressed and the paper is easy to follow.
3. The paper is well-organized and clearly written.

**Limitations:**

1. It would be interesting to discuss what are the essential differences or benefits of tree-based view learning I think it’s interesting, but more details about it should be discussed.
2. The datasets in experiments looks some old, and more compasions should be added.

**Suitability:**

3

---

### Meta-Review · Area_Chair_DqyF · 2024-06-28

**Recommendation:** Accept (Poster)
**Confidence:** 5

**Metareview:**

This paper considers the diverse and insightful inherent in multi-view data to build a tree-based view gap maintaining multi-view clustering algorithm. They conceptualize multiple views as graph structures, followed by graph pruning, which is similar to tree decision schemes. The given share-specific learning framework generates view trees for both shared and view-specific information. The authors also offered a theoretical analysis of the risks of eliminating the view gaps. Experimental results show its efficacy of the proposed method.

We have five reviewers and all of them voted for accepting this paper, due to good presentation, well organization of the whole paper, general well-designed algorithm, and sufficient experiments on multi-view learning. After rebuttal, all the reviewers are satisfied and this paper is good to be presented at ACM MM 2024.